# OpenReview forum: "Is ChatGPT a General-Purpose Natural Language Processing Task Solver?"
_EMNLP/2023/Conference — EMNLP 2023 Main_

### Official Review · Reviewer_ZytF · 2023-08-10

**Soundness:** 3

**Excitement:**

3: Ambivalent: It has merits (e.g., it reports state-of-the-art results, the idea is nice), but there are key weaknesses (e.g., it describes incremental work), and it can significantly benefit from another round of revision. However, I won't object to accepting it if my co-reviewers champion it.

**Paper Topic And Main Contributions:**

This paper investigates the zero-shot learning capabilities of ChatGPT, a large language model (LLM), by evaluating its performance on 20 popular natural language processing (NLP) datasets spanning seven task categories. While ChatGPT has gained attention for generating high-quality responses and self-correcting errors in conversations, its ability to perform a diverse range of NLP tasks without task-specific adaptation remains uncertain. The study compares ChatGPT (gpt-3.5-turbo) with the previous GPT-3.5 model (text-davinci-003) and includes zero-shot, fine-tuned, and few-shot fine-tuned results from other models.

Key findings from the research include:

ChatGPT shows potential as a generalist model across various tasks but often lags behind task-specific fine-tuned models in performance.
ChatGPT excels in tasks involving reasoning, such as arithmetic reasoning, but struggles with commonsense, symbolic, and logical reasoning tasks.
ChatGPT outperforms GPT-3.5 in natural language inference tasks and question answering tasks that require reasoning, particularly in classifying entailment.
ChatGPT performs well in dialogue tasks.
For summarization tasks, ChatGPT generates longer summaries but performs worse than GPT-3.5, and limiting summary length further harms performance.
Both ChatGPT and GPT-3.5 face challenges in tasks like sequence tagging.
ChatGPT's sentiment analysis ability surpasses that of GPT-3.5.
The evaluation involves a variety of datasets covering tasks like reasoning, natural language inference, question answering, dialogue, summarization, named entity recognition, and sentiment analysis. The study employs a two-stage prompting approach for zero-shot CoT (completion of tasks), where the model generates a rationale before generating the final answer.

In conclusion, the research highlights ChatGPT's strengths and limitations across different NLP tasks in a zero-shot setting. While it demonstrates promise as a generalist model, its performance varies across tasks, and there is room for improvement in addressing task-specific challenges. The findings offer insights for leveraging ChatGPT's capabilities in various NLP tasks and guiding future research to enhance generalist models.

**Questions For The Authors:**

It is not clear to me how you did Chain of Thoughts.

**Reasons To Accept:**

This paper is written in a good English and highlights interesting ChatGPT's strenghts and limitations, showing how his performances vary across tasks.

**Reasons To Reject:**

The content of this paper is not very clear (for example, the way authors use COT). Moreover, its core is not new to me, due to the fact that in the previous months I read many papers who cited it.

**Reproducibility:**

1: Could not reproduce the results here no matter how hard they tried.

**Reviewer Confidence:**

4: Quite sure. I tried to check the important points carefully. It's unlikely, though conceivable, that I missed something that should affect my ratings.

---

> ### Author Rebuttal · Authors · 2023-08-28
>
> Dear Reviewer:
>
> Thank you for the positive assessment and insightful comments.
>
> **Q1. The content of this paper is not very clear (for example, the way authors use COT). Moreover, its core is not new to me, due to the fact that in the previous months I read many papers who cited it. / It is not clear to me how you did Chain of Thoughts.**
>
> As mentioned in Section 3 (line 251-261), we use the same two-stage prompting as [1] for Zero-shot-CoT. Specifically, in the first stage, we adopt “Let’s think step by step.” as the instruction P1 to induce the model to generate the rationale R. In the second stage, we use the self-generated rationale R along with the original input X and the instruction P1 as the new input to guide the model to generate the final answer. A new instruction P2, e.g., “Therefore, among A through E, the answer is”, serves as the trigger sentence for extracting the answer.
>
> Besides, we would like to point out that it should not be a reason for rejection that there are many papers citing our work, which instead provides strong evidence for the impact of our work.
>
> Thanks again for your feedback. We hope that this addresses your questions.
>
> [1] Large language models are zero-shot reasoners. Kojima, Takeshi, et al. NeurIPS2022

---

### Official Review · Reviewer_uDLT · 2023-08-11

**Soundness:** 4

**Excitement:**

3: Ambivalent: It has merits (e.g., it reports state-of-the-art results, the idea is nice), but there are key weaknesses (e.g., it describes incremental work), and it can significantly benefit from another round of revision. However, I won't object to accepting it if my co-reviewers champion it.

**Paper Topic And Main Contributions:**

This paper mainly focus on the analysis and comprehensive evaluation of the  zero-shot learning ability of ChatGPT  in the context of resolving diverse NLP tasks, including but not limited to reasoning and sentiment analysis.   The authors conducted an expansive series of experiments involving 20 distinct NLP benchmarks, with the purpose of  comparing ChatGPT against previous GPT-3.5 models. Notably, a thorough  analysis of each task category is presented, allowing for a deeper understanding of the model's performance.

**Reasons To Accept:**

1. This is the first effort in conducting a comprehensive assessment of ChatGPT's zero-shot learning proficiency across diverse NLP benchmarks.
2. Some findings  are interesting and meaningful, potentially offering valuable insights for future research works within the domain of NLP applications.
3. The writing is clear and easy to follow.

**Reasons To Reject:**

1. The scope of investigated prompting techniques for zero-shot learning remains constrained. Only the approach of zero-shot cot has been taken into account. However, I find it beyond doubt that a more refined prompting approach, such as zero-shot cot with self-consistency or zero-shot cot with tool augmentation, would be worthy of consideration.
2. It appears that the performance and behavior of ChatGPT evolve over time as a result of various versions. This implies that the assessments and conclusions drawn from evaluations may likewise undergo changes.
3. Some evaluation metrics such as those used for summarization tasks, are automated in nature. This can potentially result in disparities between the outcomes of human evaluations and those generated through automated means.

**Reproducibility:**

3: Could reproduce the results with some difficulty. The settings of parameters are underspecified or subjectively determined; the training/evaluation data are not widely available.

**Reviewer Confidence:**

5: Positive that my evaluation is correct. I read the paper very carefully and I am very familiar with related work.

---

> ### Author Rebuttal · Authors · 2023-08-28
>
> Dear Reviewer:
>
> Thank you for the positive assessment and insightful comments.
>
> **Q1. The scope of investigated prompting techniques for zero-shot learning remains constrained. Only the approach of Zero-shot-CoT has been taken into account. However, I find it beyond doubt that a more refined prompting approach, such as Zero-shot-CoT with self-consistency or Zero-shot-CoT with tool augmentation, would be worthy of consideration.**
>
> Due to the cost of ChatGPT, we only conduct Zero-shot-CoT experiments in the current version. We will include more refined prompting approaches for zero-shot learning, e.g., Zero-shot-CoT with self-consistency, in the final version.
>
> Here we have evaluated the performance of ChatGPT using Zero-shot-CoT with self-consistency (5 reasoning paths) on several reasoning tasks as pilot experiments. In general, self-consistency could improve the performance of Zero-shot-CoT.
>
> |  | GSM8K (arithmetic) | AQUA-RAT (arithmetic) | CSQA (commonsense) | Last Letter (symbolic) |
> |  ----  | ----  | ----  | ----  | ----  |
> | Zero-shot-CoT only | 78.9  | 53.5 | 71.5 | 70.2 |
> | self-consistency | 80.2  | 54.7 | 72.1 | 84.6 |
>
> **Q2. It appears that the performance and behavior of ChatGPT evolve over time as a result of various versions. This implies that the assessments and conclusions drawn from evaluations may likewise undergo changes.**
>
> In the ‘Limitations’ section (the footnote on line 508), we provide the exact date of evaluation (from 06/15/2023 to 06/21/2023). We will update the results and conclusions once there is a new version of ChatGPT.
>
>
> **Q3. Some evaluation metrics such as those used for summarization tasks, are automated in nature. This can potentially result in disparities between the outcomes of human evaluations and those generated through automated means.**
>
> We agree that there might be disparities between the outcomes of human evaluations and those generated through automated means. Therefore, we conduct human evaluations to analyze the summary quality of ChatGPT and GPT-3.5. Specifically, we ask four humans (volunteer graduate students with professional English competency) to compare summaries generated by ChatGPT and GPT-3.5 on 50 randomly sampled test examples on SAMSum. The results shown below are consistent with the conclusion (ChatGPT underperforms GPT-3.5)  in Section 4.2.6.
>
> |  | ChatGPT better | GPT-3.5 better | Tie |
> |  ----  | ----  | ----  | ----  |
> | Percentage (%) | 33.5  | 51.0 | 15.5 |
>
> Thanks again for your feedback. We hope that this addresses your questions.

---

### Official Review · Reviewer_DEqi · 2023-08-11

**Soundness:** 5

**Excitement:**

3: Ambivalent: It has merits (e.g., it reports state-of-the-art results, the idea is nice), but there are key weaknesses (e.g., it describes incremental work), and it can significantly benefit from another round of revision. However, I won't object to accepting it if my co-reviewers champion it.

**Paper Topic And Main Contributions:**

This paper is about the evaluation of ChatGPT's zero-shot learning ability on 20 popular NLP datasets covering 7 representative task categories. The paper addresses the question of whether ChatGPT is a general-purpose natural language processing task solver. The main contributions of the paper are:

1. Empirical analysis of ChatGPT's zero-shot learning ability on 20 popular NLP datasets covering 7 representative task categories.
2. Identification of the effectiveness and limitations of the current version of ChatGPT in solving NLP tasks zero-shot.
3. Comparison of ChatGPT's performance on tasks that require reasoning capabilities versus tasks that require sequence tagging.
4. Discussion of the implications of the findings for the development of more effective and efficient NLP models.

The paper makes a valuable contribution to the field of NLP by providing a comprehensive evaluation of ChatGPT's zero-shot learning ability and highlighting the strengths and weaknesses of the current version of the model. The paper also provides insights into the types of tasks that are more challenging for ChatGPT and suggests directions for future research in the development of more effective and efficient NLP models.

**Reasons To Accept:**

The strengths of this paper include:

1. Comprehensive evaluation: The paper provides a comprehensive evaluation of ChatGPT's zero-shot learning ability on 20 popular NLP datasets covering 7 representative task categories. This evaluation provides valuable insights into the strengths and weaknesses of the current version of ChatGPT.

2. Empirical analysis: The paper presents an empirical analysis of ChatGPT's performance on various NLP tasks, which is a valuable contribution to the field of NLP.

3. Comparison of performance: The paper compares ChatGPT's performance on tasks that require reasoning capabilities versus tasks that require sequence tagging, which provides insights into the types of tasks that are more challenging for ChatGPT.

4. Implications for future research: The paper discusses the implications of the findings for the development of more effective and efficient NLP models, which is valuable information for researchers working in this area.

If this paper were to be presented at the conference or accepted into Findings, the main benefits to the NLP community would be:

1. Access to valuable insights: The paper provides valuable insights into the strengths and weaknesses of ChatGPT, which can help researchers to develop more effective and efficient NLP models.

2. Empirical evidence: The paper presents empirical evidence of ChatGPT's performance on various NLP tasks, which can help researchers to better understand the capabilities of this model.

3. Comparison of performance: The paper compares ChatGPT's performance on tasks that require reasoning capabilities versus tasks that require sequence tagging, which can help researchers to identify the types of tasks that are more challenging for this model.

4. Implications for future research: The paper discusses the implications of the findings for the development of more effective and efficient NLP models, which can help researchers to identify areas for future research.

**Reasons To Reject:**

According to the conference policies, weaknesses should not include issues related to writing style, formatting, or other minor issues that can be easily addressed. With that in mind, some potential weaknesses of this paper are:

1. Limited scope: The paper excludes larger-scale datasets and more task categories due to the cost of ChatGPT, which might prevent further insights.

2. Limited generalizability: The paper only evaluates ChatGPT's zero-shot learning ability and does not consider its performance on tasks that require fine-tuning or transfer learning.

3. Lack of novelty: While the paper provides a comprehensive evaluation of ChatGPT's zero-shot learning ability, it does not introduce any new models or techniques.

4. Limited comparison: The paper only compares ChatGPT's performance with existing LLMs and does not compare it with other state-of-the-art models.

The main risks of having this paper presented at the conference or accepted into Findings are:

1. Misinterpretation of results: There is a risk that the results of the paper may be misinterpreted or overgeneralized, leading to incorrect conclusions or recommendations.

2. Limited impact: The paper may have limited impact on the NLP community if it does not introduce any new models or techniques.

3. Limited interest: The paper may not generate significant interest among conference attendees or readers if it does not address a pressing research question or provide novel insights.

4. Limited relevance: The paper may not be relevant to all subfields of NLP, which could limit its appeal to a broader audience.

**Reproducibility:**

2: Would be hard pressed to reproduce the results. The contribution depends on data that are simply not available outside the author's institution or consortium; not enough details are provided.

**Reviewer Confidence:**

2: Willing to defend my evaluation, but it is fairly likely that I missed some details, didn't understand some central points, or can't be sure about the novelty of the work.

---

> ### Author Rebuttal · Authors · 2023-08-28
>
> Dear Reviewer:
>
> Thank you for the positive assessment and insightful comments.
>
> **Q1. Limited scope: The paper excludes larger-scale datasets and more task categories due to the cost of ChatGPT, which might prevent further insights. / Limited relevance: The paper may not be relevant to all subfields of NLP, which could limit its appeal to a broader audience.**
>
> We agree that conducting experiments on larger-scale datasets and more task categories can provide more useful insights, and make our work appeal to a broader audience. However, due to the cost of ChatGPT, we exclude them in the current version. We will include larger-scale datasets (such as MMLU) and more task categories (such as machine translation) in the final version.
>
> Here we have conducted additional experiments on the zero-shot performance of ChatGPT on MMLU, which is **65.6**
>
>
> **Q2. Lack of novelty: While the paper provides a comprehensive evaluation of ChatGPT's zero-shot learning ability, it does not introduce any new models or techniques. / Limited impact: The paper may have limited impact on the NLP community if it does not introduce any new models or techniques. / Limited interest: The paper may not generate significant interest among conference attendees or readers if it does not address a pressing research question or provide novel insights.**
>
> (1) We would like to point out that the main goal of our work is to extensively explore the zero-shot learning ability of ChatGPT, rather than to design new models or techniques.
>
> (2) Our work is in sharp contrast to existing work ([1] and [2]) in the following three aspects.
>
> Firstly, our work is the first to systematically explore whether ChatGPT could serve as a generalist model for various tasks.
>
> Secondly, our work is more comprehensive in terms of evaluation than existing work (preprints). For example, we evaluate the SAMSum dataset using the entire test set (819 samples) while [1] only uses 50 samples.
>
> Thirdly, our work provides valuable insights into the strengths and weaknesses of ChatGPT which could pave the road for future research.
>
>
> **Q3. Limited generalizability: The paper only evaluates ChatGPT's zero-shot learning ability and does not consider its performance on tasks that require fine-tuning or transfer learning.**
>
> As mentioned in the ‘Introduction’ section (line 96), we mainly want to evaluate whether ChatGPT can serve as a generalist model that can perform multiple tasks without learning. Therefore, we evaluate ChatGPT’s zero-shot learning ability on a diverse range of tasks.
>
> We will include more analysis of ChatGPT's few-shot in-context learning capability in the final version. Here we have conducted additional experiments on the 5-shot performance of ChatGPT on MMLU, which is **67.7**
>
>
>
> **Q4. Limited comparison: The paper only compares ChatGPT's performance with existing LLMs and does not compare it with other state-of-the-art models.**
>
> Thanks for your suggestion! We will include more state-of-the-art models, e.g., Llama 2, in the final version. Here we report the performance comparison between ChatGPT and Llama 2 on MMLU (5-shot). Llama-2-70b outperforms ChatGPT by about 2%, demonstrating the effectiveness of recent open-source LLMs.
>
> | ChatGPT (gpt-3.5-turbo)  | Llama-2-7b | Llama-2-13b | Llama-2-70b |
> |  ----  | ----  | ----  | ----  |
> | 67.7 | 46.9  | 55.8 | 69.8 |
>
>
> **Q5. Misinterpretation of results: There is a risk that the results of the paper may be misinterpreted or overgeneralized, leading to incorrect conclusions or recommendations.**
>
> As we have already included 20 NLP datasets covering 7 representative task categories (using the full test or validation set), the risk should be minimal. We will try to cover more typical datasets in the final version to make the conclusions more reliable.
>
>
> Thanks again for your feedback. We hope that this addresses your questions.
>
> [1] A multitask, multilingual, multimodal evaluation of chatgpt on reasoning, hallucination, and interactivity. Bang, Yejin, et al. arXiv preprint arXiv:2302.04023
>
> [2] ChatGPT: Jack of all trades, master of none. Kocoń, Jan, et al. Information Fusion (2023)

---

### Official Review · Reviewer_JJED · 2023-08-12

**Soundness:** 4

**Excitement:**

3: Ambivalent: It has merits (e.g., it reports state-of-the-art results, the idea is nice), but there are key weaknesses (e.g., it describes incremental work), and it can significantly benefit from another round of revision. However, I won't object to accepting it if my co-reviewers champion it.

**Missing References:**

Bang et al. (https://arxiv.org/abs/2302.10724) evaluate ChatGPT on a number of NLP datasets, including the SamSum dataset used in this paper. However, it should be noted the paper under review is more comprehensive, using the entire 819 test set samples whereas  Bang et al. only use 50.

Kocoń et al. (https://arxiv.org/abs/2302.04023) also evaluate ChatGPT on NLP tasks such as mathematical reasoning and natural language inference, though there is no overlap in datasets.

**Paper Topic And Main Contributions:**

This paper evaluates the zero-shot performance of ChatGPT on 20 NLP datasets. It seeks to understand the strengths and limitations of ChatGPT on different tasks, and compares its performance to GPT-3.5 as well as existing LLMs.

**Questions For The Authors:**

A: Since ChatGPT (gpt3.5-turbo) is being constantly updated, would it be possible to identify the exact version of ChatGPT used for these evaluations (i.e. by indicating the date of evaluation)?

EDIT: the authors provide this information in the Limitations section

**Reasons To Accept:**

1. The paper covers a variety of NLP tasks and convincingly demonstrates that ChatGPT's current zero-shot performance is outperformed by existing few-shot/fine-tuned approaches.

2. The authors explore the efficacy of zero-shot Chain-of-Thought Prompting and show it can improve or degrade performance depending on the task.

3. The authors provide qualitative studies of ChatGPT's failure cases and break down performance by class to allow for more fine-grained comparisons between models.

**Reasons To Reject:**

1. Potential lack of novelty: The paper's Related Work section does not mention existing work (preprints) that also evaluate ChatGPT on NLP tasks, including one paper where there is overlap in datasets studied. I have listed these in the Missing References section.

**Reproducibility:**

4: Could mostly reproduce the results, but there may be some variation because of sample variance or minor variations in their interpretation of the protocol or method.

**Reviewer Confidence:**

3: Pretty sure, but there's a chance I missed something. Although I have a good feel for this area in general, I did not carefully check the paper's details, e.g., the math, experimental design, or novelty.

---

> ### Author Rebuttal · Authors · 2023-08-28
>
> Dear Reviewer:
>
> Thank you for the positive assessment and insightful comments.
>
> **Q1. Potential lack of novelty: The paper's Related Work section does not mention existing work (preprints) that also evaluate ChatGPT on NLP tasks, including one paper where there is overlap in datasets studied. I have listed these in the Missing References section.**
>
> Thanks for your suggestions! We will cite [1] and [2] as related work in the final version.
>
> Our work is in sharp contrast to [1] and [2] in the following three aspects.
>
> Firstly, our work is the first to systematically explore whether ChatGPT could serve as a generalist model for various tasks.
>
> Secondly, as you mentioned, our work is more comprehensive in terms of evaluation than existing work (preprints).
>
> Thirdly, our work provides valuable insights into the strengths and weaknesses of ChatGPT which could pave the road for future research.
>
>
> **Q2. Since ChatGPT (gpt3.5-turbo) is being constantly updated, would it be possible to identify the exact version of ChatGPT used for these evaluations (i.e. by indicating the date of evaluation)?**
>
> We do provide the exact date of evaluation (from 06/15/2023 to 06/21/2023) in the ‘Limitations’ section (the footnote on line 508). We will update the results and conclusions once there is a new version of ChatGPT.
>
>
> Thanks again for your feedback. We hope that this addresses your questions.
>
>
> [1] A multitask, multilingual, multimodal evaluation of chatgpt on reasoning, hallucination, and interactivity. Bang, Yejin, et al. arXiv preprint arXiv:2302.04023
>
> [2] ChatGPT: Jack of all trades, master of none. Kocoń, Jan, et al. Information Fusion (2023)

---

### Meta-Review · Area_Chair_nPNH · 2023-09-17

**Recommendation:** 4

**Metareview:**

This paper evaluates the zero-shot performance of ChatGPT on 20 NLP datasets. It seeks to understand the strengths and limitations of ChatGPT on different tasks, and compares its performance to GPT-3.5 as well as existing LLMs.

Reasons To Accept:
- The paper covers a variety of NLP tasks and convincingly demonstrates that ChatGPT's current zero-shot performance is outperformed by existing few-shot/fine-tuned approaches.
- The authors explore the efficacy of zero-shot Chain-of-Thought Prompting and show it can improve or degrade performance depending on the task
- Comparison of performance: The paper compares ChatGPT's performance on tasks that require reasoning capabilities versus tasks that require sequence tagging, which provides insights into the types of tasks that are more challenging for ChatGPT.
- Implications for future research: The paper discusses the implications of the findings for the development of more effective and efficient NLP models, which is valuable information for researchers working in this area.


Reasons To Reject:
- Potential lack of novelty: The paper's Related Work section does not mention existing work (preprints) that also evaluate ChatGPT on NLP tasks, including one paper where there is overlap in datasets studied.
- Limited scope: The paper excludes larger-scale datasets and more task categories due to the cost of ChatGPT, which might prevent further insights.
- Limited generalizability: The paper only evaluates ChatGPT's zero-shot learning ability and does not consider its performance on tasks that require fine-tuning or transfer learning.
- Lack of novelty: While the paper provides a comprehensive evaluation of ChatGPT's zero-shot learning ability, it does not introduce any new models or techniques.
- Limited comparison: The paper only compares ChatGPT's performance with existing LLMs and does not compare it with other state-of-the-art models.

---

### Decision · Program_Chairs · 2023-10-07

**Decision:**

Accept-Main

**Comment:**

This paper evaluates the zero-shot performance of ChatGPT on 20 NLP datasets. It seeks to understand the strengths and limitations of ChatGPT on different tasks, and compares its performance to GPT-3.5 as well as existing LLMs.

Reasons To Accept:
- The paper covers a variety of NLP tasks and convincingly demonstrates that ChatGPT's current zero-shot performance is outperformed by existing few-shot/fine-tuned approaches.
- The authors explore the efficacy of zero-shot Chain-of-Thought Prompting and show it can improve or degrade performance depending on the task
- Comparison of performance: The paper compares ChatGPT's performance on tasks that require reasoning capabilities versus tasks that require sequence tagging, which provides insights into the types of tasks that are more challenging for ChatGPT.
- Implications for future research: The paper discusses the implications of the findings for the development of more effective and efficient NLP models, which is valuable information for researchers working in this area.


Reasons To Reject:
- Potential lack of novelty: The paper's Related Work section does not mention existing work (preprints) that also evaluate ChatGPT on NLP tasks, including one paper where there is overlap in datasets studied.
- Limited scope: The paper excludes larger-scale datasets and more task categories due to the cost of ChatGPT, which might prevent further insights.
- Limited generalizability: The paper only evaluates ChatGPT's zero-shot learning ability and does not consider its performance on tasks that require fine-tuning or transfer learning.
- Lack of novelty: While the paper provides a comprehensive evaluation of ChatGPT's zero-shot learning ability, it does not introduce any new models or techniques.
- Limited comparison: The paper only compares ChatGPT's performance with existing LLMs and does not compare it with other state-of-the-art models.